# Green R&D Financing Strategy in Platform Supply Chain with Data-Driven Marketing

**Yanfei Xia, Quan Guo, Hao Sun, Ke Li and Zongyu Mu *** 

Department of Management Science and Engineering, School of Business, Qingdao University, Qingdao 266071, China; xiayanfei2020@163.com (Y.X.); guoquan0015@163.com (Q.G.); sunhao@qdu.edu.cn (H.S.); like920418@163.com (K.L.)
* Correspondence: mzydragon@163.com; Tel.: +86-153-7689-2910

**Abstract:** Platform enterprises can improve green R&D efficiency by data-driven marketing (DDM) activities and can also provide financing assistance to manufacturers. In this context, for a platform supply chain consisting of one manufacturer facing a shortage of green R&D funds and a one third-party platform, this paper develops four game models under two financing channels (bank financing channel and platform financing channel) and two selling modes (agency selling mode and reselling mode). The equilibrium results of different models are derived and compared, and then the choices of selling mode and financing channel from the perspectives of both the manufacturer and the platform are analyzed. The conclusions show that the consumers' sensitivities to green R&D and DDM activities, as well as service commission fee, are major factors influencing green R&D level and both parties' choice of selling mode and financing channel. In most cases, a platform financing channel can promote the green R&D level better and is more beneficial to the manufacturer and the platform. Only in a few cases, the two parties prefer the reselling mode and bank financing channel. However, agent selling with bank financing will never be their optimal strategy. There exists four situations in which the manufacturer and the platform can agree on a same strategy on selling mode and financing channel.

**Keywords:** green R&D; financing; platform supply chain; data-driven marketing

## 1. Introduction

The rapid development of the global economy has caused serious environmental pollution. Therefore, many enterprises carry out green R&D, production, and operation management activities in the supply chain [1]. For example, Haier Bio fulfills corporate responsibility via R&D, and carries the concept of green and low-carbon development throughout the whole chain of production and operation [2]. Moreover, Haier has reached agreement with JD.com to promote products through data-driven marketing technology. In April 2021, Honeywell made a solemn commitment to sustainability by achieving carbon neutrality in all of its operations and facilities by 2035, and has repeatedly set and exceeded aggressive sustainable development goals, reducing greenhouse gas emissions from its operations and facilities by more than 90% since 2004 [3]. The "Green 360" program initiated by Wal-Mart states that all the products it sells must be 100% zero-cost manufacturing and energy renewable [4], and Wal-Mart can provide financing loans to these businesses.

However, green R&D needs a huge amount of capital, and it is often far from enough to rely only on manufacturers themselves when they conduct product R&D activities and expand new fields. Thus, how to relieve the shortage of capital in developing green products becomes one of the biggest challenges for many enterprises worldwide. Currently, banking financing is a main financing channel. Many banking institutions have started lending to green SEMs since the International Finance Corporation (IFC) announced the equator Principles in 2003. China has implemented green credit policies and encouraged

banks and other financial institutions to provide credit support to cash-strapped, low-carbon enterprises since 2007 [5]. By the end of 2018, the accumulated amount of green credit provided by the industrial and Commercial Bank of China, Agricultural Bank of China, China Construction Bank, Bank of China, Bank of Communications, and Postal Savings Bank of China had exceeded CNY 4.4 trillion [6]. Cornerstone Technologies Holdings Limited announced a green loan financing of about HKD 150 million in 2022. All these measures have played a positive role in supporting green R&D for the manufacturers. However, as capital providers, banks usually decide whether or not to grant loans by assessing the risks of lending enterprises only, albeit without considering supply chain risks, i.e., the risks of upstream and downstream stakeholders of enterprises. This is not conducive to the sustainable development of supply chain members.

With the rapid progress of Internet technology, the platform economy is conducive to improving the allocation efficiency of social resources and giving birth to many new forms of business with the characteristics of networking, digitalization, and intelligence. For example, JD.com collaborates with brands such as Huawei, Coach, and Burberry under the reselling model, in which the brands sell their products to JD.com at wholesale prices, and then JD.com sells them to consumers at a markup. However, when JD.com cooperates with Topsports, Sephora, and other brands, it adopts the agency selling model, and charges certain platform commission fees and sells its products through JD platform. Since 2010, the Amazon platform in the United States has not only sold various enterprises' products through agency selling mode, but also opened a new selling channel to sell products. In addition, the platform provides enterprises with a new way of financing besides financing by bank, that is, to provide loans to manufacturers or suppliers who sell products on its platform. Thus, the platforms can not only provide online distribution channels but also online financing services for enterprises through supplier selling, credit and other transaction data, so as to facilitate the distribution of enterprises and relieve their capital pressure. For example, Alibaba, the owner of the Taobao platform, provides loans to B2B enterprises operating on its platform [7]. In 2018, JD.com launched "JD Express Bank", a new online financial service based on big data analysis and new Internet technologies. Moreover, JD.com has also established the "Jingdong Bao" financing channel to meet the needs of small and median enterprises. Companies selling goods on Amazon platform can obtain loans from the Amazon platform, with the loan amount ranging from USD 1000 to 750,000 up to one year, and the annual interest rate ranging from 6% to 17% [8]. Platform financing channel has become a more convenient and efficient way: on one hand, it is conducive to the production and operation activities of enterprises; on the other hand, it can also make the platform obtain a higher income. Furthermore, the platform has the natural advantage of data collection, which can describe, predict, analyze, and guide consumers' purchasing behaviors based on the acquired data and provide high-quality and accurate marketing services for all kinds of products, including green products [9]. At the same time, the platform's data-driven marketing (DDM) activities can promote green products, strengthen green consumption concept of consumers, and improve the market share of green products [10]. Therefore, more and more manufacturers are selling green products on the platform. For example, JD.com uses DDM to cooperate with many well-known brands in the industry, such as mobile phone brand Huawei and food and home appliance brand Haier to sell their green products.

As mentioned above, under the background of platform marketing, when the manufacturers are faced with a shortage of funds during green products production and R&D innovation, it has alternative financing channels and selling modes to choose from. Thus, this paper focuses on the choices of selling mode and financing channel in a green platform supply chain. In specific, we try to answer the following questions:

1.  What factors affect the levels of green R&D and DDM activities and then affect all members' operational decisions and profits in the platform supply chain with the manufacturer's green R&D capital-constrained?

2.　　Under an agency selling or reselling mode, which financing channel is more conducive to improve the level of green R&D and the profits of the manufacturer and the platform?

3.　　Under the combination of different selling modes and financing channels, which strategy is more conducive to improving operational efficiency? Can the manufacturer and the platform come to an agreement on selling mode and financing channel?

To answer the above questions, we model a platform supply chain composed of a capital-constrained manufacturer with green R&D and a platform with DDM activities, where the manufacturer can obtain capital from a bank or third-party platform. The manufacturer invests in green R&D to meet consumers' green preferences, and then an agency sells or resells green products through the platform. The platform invests in DDM to expand the sales of green products. Then, we compare the manufacturer's and the platform's profits and the level of green R&D through agency selling and reselling modes under the platform financing channel and bank financing channel, respectively. The results show that, in most cases, platform financing channel can promote the green R&D level better and is more beneficial to the manufacturer and the platform. Only in a few cases did the two parties prefer to employ the reselling mode and bank financing channel. However, agent selling with bank financing will never be their optimal strategy.

This paper makes contributions in the following two aspects. First of all, we study for the first time the impact of green R&D behavior of the manufacturer and DDM activities of the platform on the operations performance of a supply chain with an online platform that provides two selling modes for green products. Therefore, we build a link between the green supply chain and the platform supply chain with DDM activities. Second, we establish two-stage game models to explore the platform supply chain operation strategies under four different scenarios, with consideration of alternative selling modes and financing channels.

The rest of this paper is organized as follows. The relevant literature is reviewed in Section 2. Section 3 describes model background, establishes the market demand functions, and clearly presents the hypotheses and symbolic connotation. Four game models are given in Sections 4 and 5 to analyze the impact of the financing channel on the choice of selling mode. In Section 6, we verify the correctness of the results and further explore management enlightenment through numerical analysis. Section 7 summarizes the research results of this paper and presents possible further research directions in the future.

## 2. Literature Review

Three fields of achievements are relevant to our work: green supply chain management, supply chain financing management and platform supply chain management.

### 2.1. Green Supply Chain Management

Green supply chain, also known as environment-conscious supply chain, is a mode of balancing economic growth and environmental protection and incorporating green environmental behavior into enterprise management [11]. In the 1990s, scholars began to study green supply chain management. In 1994, Moffat et al. [12] proposed the concept of green procurement when they studied the impact of products on the environment and advised that companies choose the right raw materials through environmental guidelines and focus on recycling. In 1996, scholars from Michigan State University proposed the embryonic form of green supply chain management theory. On this basis, Handfield [13] broadens the meaning of environmentally responsible manufacturing for enterprises, providing a reference for enterprises in the supply chain to reduce their impact on the environment in the process of operation.

In recent years, many scholars have put forward many insightful views on the concept of green supply chain and its management. Some scholars analyze green supply chain management from the aspects of green procurement, environmental innovation concept, product manufacturing and recycling, environmental benefits [14–18], and so on. From the perspective of e-commerce and green production, Rahmani and Yavari [19] define green

supply chain management as that which considers the necessity of the environment in supply chain management while paying attention to ecological benefits in product design, material supply, processing, transportation, and product recycling and reuse interaction. De et al. [20], taking a Norwegian Salmon Supply Chain Network as evidence, studied the impact of different supply chain arrangements in the food supply chain on the cost of carbon emissions (a kind of green R&D cost) to solve environmental problems, and considered the restrictions related to carbon emissions. Gao et al. [21] consider the issue of green supply chain management with green standards formulated by the government and focused on the study of green products with two different green technologies. The study found that the improvement of green R&D technologies can continuously improve the environmental benefits of development-intensive green products. Ma et al. [22] found that green supply chain management is an environmental management mode to attract consumers to green consumption through manufacturers' green emission reduction technologies, which thus alleviates the global environmental crisis. Liu et al.'s [23] study from the perspective of agriculture found that green supply chain management is an effective measure to drive economic development, benefit the people, and reduce carbon emission pollution under the constraints of carbon tax. The above scholars define their understanding of green supply chain management from different industries and perspectives. By summarizing their common points, it can be seen that green supply chain management is a modern management mode that comprehensively considers environmental impact and resource efficiency. As an important way for manufacturing enterprises to fulfill policy requirements and environmental responsibilities and achieve sustainable development, green R&D has long attracted extensive attention of scholars. How to enhance the green R&D ability of enterprises in the supply chain is one of the core issues. However, the existing literature ignores the fact that most supply chain members face capital constraints when investing in green technologies; however, in reality, they are more vulnerable to capital constraints in operational decisions. Therefore, we consider whether the establishment of a financing relationship with the platform is more beneficial for manufacturers to produce green products compared to borrowing from banks during supply chain operations.

### 2.2. Supply Chain Financing Management

Traditionally, to mitigate the negative impact of capital constraints, many enterprises in the supply chain have turned to external institutions to meet their financing needs [24], among which financing by bank is one of the most common financing strategies. Some of the early literature studied co-financing and inventory decisions by incorporating capital constraints into traditional newsvendor models. Although the traditional bank credit financing may solve the financing problems of many enterprises, it is often difficult for enterprises to obtain loans directly from banks and other financial institutions, due to the lack of certain tangible internal resources, high transaction costs, high operational risks, serious information asymmetry, and high bank loan costs [25]. In addition to a bank, manufacturers can also finance through suppliers, retailers, etc. Kouvelis and Zhao [26] studied the situation where both retailers and suppliers are constrained by capital and need short-term financing, concluding that, if an optimally structured trade credit contract is provided, retailers always prefer supplier financing rather than bank financing. Wu et al. [27] studied the situation where manufacturers take measures to reduce carbon emissions, while retailers are funded by banks or manufacturers due to limited funds, and explained how financing and the capital of retailers affect order quantity and carbon emission reduction. Jin et al. [8] study three financing strategies of bilateral supply chains in which both suppliers and retailers are subject to financial constraints. The three financing strategies are, respectively, non-cooperative bank sole financing strategy, cooperative bank trade credit financing strategy, and bank supplier guarantee financing strategy. Yang et al. [28] studied financing and pricing decisions in a supply chain, and the results show that supply chain provides decision-making options for new entrants and manufacturers must choose financing by retailer or bank according to the conditions. Huang et al. [29] studied the

case of financing by bank, and the government can subsidize banks by providing financing as well as manufacturers and retailers. Fang and Xu [30] studied the green supply chain financing system composed of a capital-constrained manufacturer, a retailer, and a bank, and analyzed the financing behavior of manufacturers. Tang et al. [31] studied two innovative financing schemes emerging in recent years. The first is purchase order financing, which allows financial institutions to provide loans to suppliers by considering the value of purchase orders issued by creditworthy buyers. The second is buyer's direct financing, that is, manufacturers issue purchase contracts and loans directly to suppliers. Both of these financing methods enable suppliers to obtain production financing.

With the development of the online marketplace, the platform provides loan services for members of the functional supply chain. Platform is an emerging Internet financial business model, which has just entered the development period, but the effect of financing by platform cannot be underestimated. As a product of the Internet era, platform plays an increasingly prominent role in the financial field due to its advantages of low financing threshold and convenient financing methods, which helps to solve the financing problems of cash-strapped enterprises. Therefore, financing by platform has become a new direction of financing development. Gao et al. [32] studied how the service commission rates for the platform affect manufacturer's wholesale price decisions and retailer's order quantity decision when the retailer or manufacturer borrows money from the online P2P lending platform under the condition that both suppliers and retailers are faced with capital constraints. Yan et al. [33] studied the form of online finance and encouraged SME manufacturers, especially those with capital constraints, to consider online channel options and implement dual-channel strategies. Yan et al. [34] discussed a two-channel supply chain structure, in which retailers and e-commerce platforms can free ride on each other's selling efforts, while e-commerce platforms can provide online financial services to capital-constraint suppliers.

The issue how the manufacturer selects its financing channel has been studied by many scholars. Firstly, Dong et al. [35] showed that the financial providers under online and offline environments are different. For online supply chain finance, e-commerce platform is the financial service provider, while offline supply chain finance is provided by commercial banks or other financial institutions. Secondly, Gong et al. [36] shows that e-commerce platforms play a role in product distribution and financial provision, which will influence the decisions of more supply chain members. He believes that e-commerce platforms play an important role in the platform economy, providing distribution channels for SME manufacturers, helping them promote the distribution and production, and alleviating capital strains through online financing projects. Zhen et al. [37] established a model in which a capital-constrained manufacturer sells products through retailers and third-party platforms and can obtain financing from third-party platforms, retailers, or banks. For the manufacturer, financing from a third-party platform strategy is always better than financing from bank strategy. In conclusion, it is an open question how the dual roles of third-party platform lending and participation in direct channels affect dual-channel management of manufacturers. Third-party platforms must balance interest income with revenue-sharing payments from manufacturers when setting interest rates. Higher interest rates may increase loan revenues but reduce the number of products sold on the platform.

The above research mainly focuses on financing behavior in the general product supply chain. This paper, for the first time, studies the influence of green R&D and DDM activities on channel selection and cooperation strategy when the platform provides two selling modes of agency selling and reselling, adding new contents to the existing supply chain financing theories.

### 2.3. Platform Supply Chain Management

The Internet can provide a new way of communication for enterprises and consumers and can provide new distribution channels for manufacturers and retailers. Terry et al. [38] proposed that the platform model of online shopping could provide manufacturers with

an opportunity to expand the market. Mantin et al. [39] proposed that retailers are increasingly adopting the dual distribution channel model, that is, these retailers not only act as traditional merchants (buying and reselling goods), but also provide an online platform for third-party sellers to compete for the same customers. Li et al. [40] pointed out that retailers can gain more profits through the online and offline selling of product classification. The above research shows that manufacturers and retailers can gain benefits by choosing appropriate platform selling strategies.

At present, the two basic selling modes widely used by large e-commerce platforms are the traditional reselling mode and the new agency selling mode [41]. Many scholars have studied how manufacturers and platforms choose and innovate selling modes. Abhishek et al. [42] studied the balance of two e-retailers under different selling mode configurations when a manufacturer sells through two e-retailers, and used a stylized theoretical model to answer when e-retailers should use the agent mode and when they should use the reselling mode. Geng et al. [43] discussed the interaction between the pricing of additional products by upstream manufacturers and the selection of a selling mode for a downstream online platform, and found that the choice of selling mode affects the manufacturer's choice between additional pricing and bundling. Under the condition that the commission rate of the platform is not too low and the market potential of additional products is not too great, the platform is more inclined to the agency selling mode. Tan and Carrillo [44] explored the selling strategy choice between agency selling and wholesale (reselling) mode when digital goods are sold on an online platform and showed that agency selling mode is beneficial to the sales of digital products for the behavior of revenue-sharing and the direct control of price by an upstream publisher. Hao and Fan [45] paid attention to the pricing of e-books and e-book readers under wholesale (reselling) or agency selling mode, and they pointed out that the price of e-books in reselling mode is low because of the complementary relationship between e-books and e-book readers in the market. Liu et al. [10] investigated a platform's preference between agency selling and reselling, taking into account the influence of DDM, and established and compared four modes: NO-DDM + agency selling, NO-DDM + reselling, DDM + agency selling and DDM + reselling. The results showed that, with the improvement of DDM efficiency, the platform is more willing to adopt the reselling model. Ha et al. [46] studied selling strategy choice in a platform supply chain in which the third-party platform can provide an agency selling channel, reselling channel, or both selling channels and derive the equilibrium selling strategy choice.

In the context of high-quality economic and social development, platform economy helps enterprises achieve green transformation development. Du et al. [47] took the emerging issue of platform-oriented green advertising in practice as the research object, and discussed the role of platform economy in the development of green economy, concluding that the platform can gain more profits by using the best promotion strategy than the performance-based promotion strategy. The research on platform with DDM technology has become a hot topic for researchers recently. Traditional green modular design has the risk of losing the use of product platform planning strategy. Liu et al. [48] constructed a theoretical model of the information adoption behavior of green agricultural products in an e-commerce platform, and the results show that a safe environment and the information technology of the platform have a positive impact on consumers' willingness to receive green agricultural product information on the commercial platform. Li [49] studied how the corresponding technology cost invested by the smart platform reduces the channel cost in the smart platform supply chain, and the research shows that the platform could also invest in the smart platform technology without bearing any channel cost. However, channel structure and green R&D are not addressed in the above literature. This paper fills the gap of existing literature by analyzing the influences of DDM and green R&D on supply chain selling mode and financing channel.

### 2.4. Research Gaps

From the literature review above, we find that scholars have made great achievements in the green R&D in supply chain field and platform supply chain management, respectively. These achievements highlight the benefits of green innovation and e-commerce.

Table 1 lists the related papers and presents a summary of our work; among them, the following three papers are closely related to our research. Liu et al. [10] consider the case that the platform provides two selling modes and DDM technology at the same time; however, they assume the manufacturer produces ordinary products but not green products, and also do not involve the financing channel choice of the manufacturer. Du et al. [44] consider that the manufacturer sells green products via the platform, and the platform utilizes strategies to promote the sales of green products. However, their study is limited to agency selling provided by the platform and fails to examine the selection of selling mode when the platform may still adopt a reselling mode besides agency selling. Zhen et al. [37] compare three financing channels (third-party platform, retailer, and bank) for a capital-constrained manufacturer under an agent selling mode. Similarly, they do not take into account the platform's different selling modes with DDM techniques, and the green production is also not involved. To sum up, there is little literature on green R&D issues and DDM activities affecting the platform supply chain when the manufacturer faces capital constraints. Moreover, there is also no research on how the platform can affect the manufacturer's choices of selling mode and financing channel when it provides both agency selling and reselling selling modes and can finance for the manufacturer. To bridge this gap, we investigate a platform supply chain composed of a capital-constrained manufacturer and a platform that can provide a financing service and two selling modes, and analyze how consumers' sensitivities to green R&D and DDM activities affect the choices of selling mode and financing channel.

**Table 1.** Articles related to our study.

| Researchers | Platform Reselling (R)/Agency Selling(A) | Green Supply Chain | DDM | Financing by Bank | Financing by Platform |
|---|---|---|---|---|---|
| Rahmani and Yavari [19] (2018) | | √ | | | |
| Gao et al. [21] (2020) | | √ | | | |
| Ma et al. [22] (2021) | | √ | | | |
| Liu et al. [23] (2021) | | √ | | | |
| Fang and Xu [30] (2020) | | √ | | √ | |
| Gao et al. [32] (2018) | A | | | | √ |
| Zhen et al. [37] (2020) | A | | | √ | √ |
| Abhishek et al. [42] (2016) | A/R | | | | |
| Du et al. [47] (2019) | A | √ | | | √ |
| Liu et al. [48] (2020) | A/R | | √ | | |
| Our Work | A/R | √ | √ | √ | √ |

## 3. Model Development

For a platform supply chain consisting of a capital-constrained manufacturer and an online platform, the manufacturer invests in green R&D to produce green products and sells them through the platform with DDM activities. The online platform can provide the manufacturer with two selling modes: reselling mode and agency selling mode. In the reselling mode, the platform, acting as retailer, buys products from the manufacturer at a certain wholesale price and then sells them to consumers at a markup; in the agency selling mode, the platform displays the manufacturer's products, and consumers buy products directly from the manufacturer through the platform. After the manufacturer sells the products to consumers, the platform charges the manufacturer a certain service commission fee for each unit of sold products, such as Tmall, JD.com, Amazon, and other platforms. In addition, the platform can also provide a financing service for the manufacturer. Thus, the manufacturer can make a choice between a bank financing channel and a platform financing channel.

Based on the above analysis, there may exist four decision models for the platform supply chain, namely, agency selling with bank financing, reselling with bank financing, agency selling with platform financing, and reselling with platform financing, as shown in Figure 1.

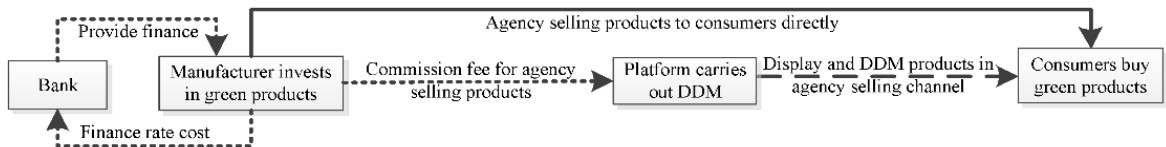

**Model 1** Platform supply chain of green R&D with agency selling mode and financing by bank

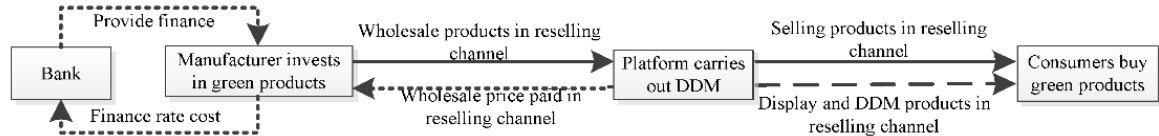

**Model 2** Platform supply chain of green R&D with reselling strategy and financing by bank

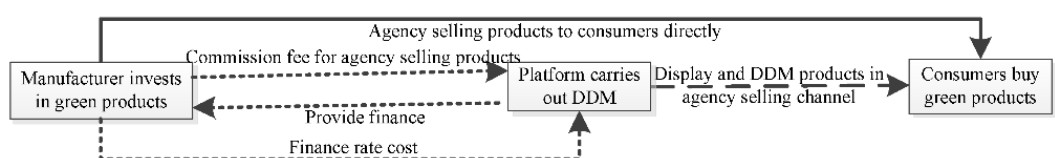

**Model 3** Platform supply chain of green R&D with agency selling strategy and financing by platform

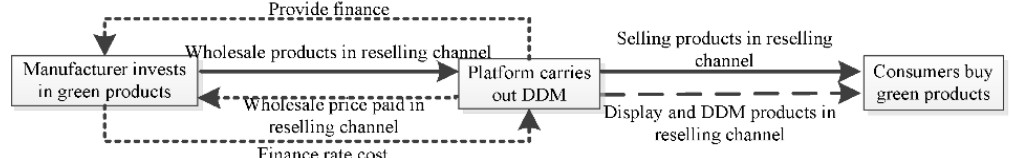

**Model 4** Platform supply chain of green R&D with reselling strategy and financing by platform

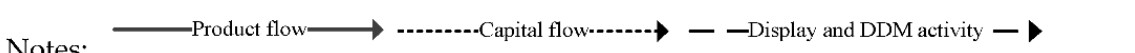

Notes:

**Figure 1.** Operation diagram of the platform supply chain under the constraint of green R&D capitals.

**Assumption 1.** *There is a Stackelberg game relationship between the manufacturer and the platform. Both the two parties are completely rational and have information symmetry with each other.*

**Assumption 2.** *The manufacturer determines the level of green R&D (e) of products. The higher the level of product green R&D, the higher the cost of investment. Since the investment cost of new technology usually exhibits diseconomies of scale, it is assumed that the investment in improving the level of green R&D is expressed as $\eta_e e^2/2$. $\eta_e$ is the sensitivity coefficient of green R&D investment cost, and $\eta_e$ can be normalized to 1 on the premise that the model is easy for quantitative analysis while the management insight is unchanged; Xia and Niu [50] have taken a similar approach to the issue of green R&D in the supply chain.*

**Assumption 3.** *The platform can provide an agency selling and reselling strategy to the manufacturer. Under the agency selling strategy, the manufacturer directly sells green products through the platform and sets the unit selling price of products as p. The platform charges a service commission fee u for each unit of product sold by the manufacturer, thus the total product service commission*

*fee paid by a manufacturer to the platform is uD. In practice, 1351 auto parts through an online mall platform will charge service commission fees to products sold by the manufacturers. Under the reselling strategy, the manufacturer sells the green products to the platform at a wholesale price w, and then the platform sells the products at a price p to consumers with a premium of m, i.e., p = w + m [51]. For the products selling by platform, the platform uses the acquired consumer behavior information to carry out DDM activities. Moreover, we set the level of DDM as t. Since the investment cost of new technology is usually characterized by diseconomies of scale, the DDM cost of the platform with a DDM level t is $\eta_t t^2/2$. $\eta_t$ is the sensitivity coefficient of DDM investment. It can be normalized into 1 on the premise that the management insights remain unchanged. Xia and Niu [50] adopted a similar approach when they studied the marketing activities of members in the supply chain.*

**Assumption 4.** *The green R&D implemented by the manufacturer can enhance the perception quality of consumers with green preferences in the market. Moreover, the DDM activities carried out by the platform can also help consumers better understand the products and raise their perception quality of the products. Therefore, both green R&D and DDM activities can increase the market demand for products. Following Agi and Yan [52] and Zheng et al. [53], we assume the demand function of consumers meets the uniform distribution of [0, 1], and then the consumer surplus is $V = \mu - p + \alpha e + \beta t$. When consumer surplus is greater than 0, i.e., $\mu > p - \alpha e - \beta t$, consumers buy products, and the market demand function of green preference consumers for green products can be calculated as $D = \int_{p-\alpha e-\beta t}^{1} 1 du = 1 - p + \alpha e + \beta t$. In this equation, 1 is the potential market size, $\alpha$ is the consumers' sensitivity to the green R&D level e, and $\beta$ is the consumers' sensitivity to the DDM level t.*

**Assumption 5.** *The manufacturer faces a shortage of capital to carry out green R&D activities and needs to obtain funds from the bank or the platform. The financing rate is $\gamma$, therefore the interest paid by the manufacturer to the bank or platform is $\gamma e^2/2$.*

The subscript $i \in \{M, P\}$ represents the manufacturer and the platform, respectively. The superscript $j \in \{a, r, bf, pf\}$ represents the agency selling mode, reselling mode, bank financing channel, and platform financing channel, respectively. The superscript * indicates equilibrium solutions or profits. Detailed symbols and meanings are shown in Table 2.

**Table 2.** Symbol description.

| Symbol | Meanings |
| --- | --- |
| $D$ | the market demand function of green products |
| $w$ | the wholesale price of green products |
| $m$ | the markup price of green products |
| $p$ | the unit selling price of green products |
| $e$ | the level of green R&D |
| $t$ | the level of DDM activities |
| $\gamma$ | the interest rate |
| $\alpha$ | the consumers' sensitivity to the green R&D level |
| $\beta$ | the consumers' sensitivity to the DDM level |
| $u$ | unit service commission fee charged by the platform |

## 4. Financing by Bank Model

### 4.1. Agency Selling Model

Under the agency selling strategy, when the manufacturer obtains financing from the bank, the bank decides the financing interest rate first, then the manufacturer decides the product selling price and the green R&D level, and finally the platform decides data-driven marketing technology level.

The profit functions of manufacturer, platform, and bank are as follows:

$$\Pi_M^{a-bf}(p,e) = (p-u)D - (1+\gamma)\frac{e^2}{2} \tag{1}$$

$$\Pi_P^{a-bf}(t) = uD - \frac{t^2}{2} \tag{2}$$

$$\Pi_B^{a-bf}(\gamma) = \frac{\gamma e^2}{2} \tag{3}$$

**Proposition 1.** *The equilibrium decisions of agency selling strategy financing by bank of platform supply chain with green R&D is:* $D^{a-bf*} = \frac{(4-\alpha^2)[1-u(1-\beta^2)]}{4(2-\alpha^2)}$, $p^{a-bf*} = \frac{(4-\alpha^2)[1+u(1+\beta^2)]-2\alpha^2 u}{4(2-\alpha^2)}$, $e^{a-bf*} = \frac{\alpha[1-u(1-\beta^2)]}{2(2-\alpha^2)}$, $t^{a-bf*} = \beta u$, $\gamma^{a-bf*} = \frac{(2-\alpha^2)}{2}$, $\Pi_M^{a-bf*} = \frac{(4-\alpha^2)[1-u(1-\beta^2)]^2}{8(2-\alpha^2)}$, $\Pi_P^{a-bf*} = \frac{u[\alpha^2\beta^2 u+(4-\alpha^2)(1-u)]}{4(2-\alpha^2)}$, $\Pi_B^{a-bf*} = \frac{\{\alpha[1-u(1-\beta^2)]\}^2}{16(2-\alpha^2)}$.

The proof is given in Appendix A.

**Conclusion 1.** (*a*) $\frac{\partial D^{a-bf*}}{\partial\alpha} > 0, \frac{\partial p^{a-bf*}}{\partial\alpha} > 0, \frac{\partial e^{a-bf*}}{\partial\alpha} > 0, \frac{\partial t^{a-bf*}}{\partial\alpha} = 0, \frac{\partial\gamma^{a-bf*}}{\partial\alpha} < 0, \frac{\partial\Pi_M^{a-bf*}}{\partial\alpha} > 0, \frac{\partial\Pi_P^{a-bf*}}{\partial\alpha} > 0, \frac{\partial\Pi_B^{a-bf*}}{\partial\alpha} > 0.$ (*b*) $\frac{\partial D^{a-bf*}}{\partial\beta} > 0, \frac{\partial p^{a-bf*}}{\partial\beta} > 0, \frac{\partial e^{a-bf*}}{\partial\beta} > 0, \frac{\partial t^{a-bf*}}{\partial\beta} > 0, \frac{\partial\gamma^{a-bf*}}{\partial\beta} = 0, \frac{\partial\Pi_M^{a-bf*}}{\partial\beta} > 0, \frac{\partial\Pi_P^{a-bf*}}{\partial\beta} > 0, \frac{\partial\Pi_B^{a-bf*}}{\partial\beta} > 0.$ (*c*) $\frac{\partial D^{a-bf*}}{\partial u} < 0, \frac{\partial p^{a-bf*}}{\partial u} > 0, \frac{\partial e^{a-bf*}}{\partial u} < 0, \frac{\partial t^{a-bf*}}{\partial u} > 0, \frac{\partial\gamma^{a-bf*}}{\partial u} = 0, \frac{\partial\Pi_M^{a-bf*}}{\partial u} < 0, \frac{\partial\Pi_P^{a-bf*}}{\partial u} > 0$ *when* $u < \frac{(4-\alpha^2)}{2[4-\alpha^2(1+\beta^2)]}, \frac{\partial\Pi_P^{a-bf*}}{\partial u} \leq 0$ *when* $u \geq \frac{(4-\alpha^2)}{2[4-\alpha^2(1+\beta^2)]}, \frac{\partial\Pi_B^{a-bf*}}{\partial u} < 0.$

Conclusion 1 shows that, in case of agent selling strategy financing from the bank, when the consumers' sensitivity to the level of green R&D increases, the number of consumers willing to buy green products increases according to the market demand function of products, and the manufacturer will increase the unit selling price of green products. In addition, the manufacturer is also encouraged to improve green R&D level, so that more potential consumers are willing to buy green products. The interest rate by bank decreases with the increase in the consumers' sensitivity to green R&D. At this point, the increase in market demand for green products, the increase in unit selling price of green products, and the decrease in interest rate during financing from bank brings more revenue to the manufacturer than an increase in the cost of green R&D, therefore the profit of the manufacturer increases. However, the consumers' sensitivity to green R&D does not affect the platform's investment in DDM activities. Thus, as the market demand for green products increases, the profit obtained by the platform increases. The bank's profit depends both on the green R&D level and the financing interest rate. When the consumers' sensitivity to green R&D increases a reduction in the financing interest rate through a bank, it makes the manufacturer more interested in investing in green products. At this time, an increase in green R&D level can make up for the reduction in financing interest rate, so that the bank's profit also increases.

An increase in the consumers' sensitivity to the level of DDM activities can inspire the platform to raise the level of DDM activities and motivate the manufacturer to invest more in green R&D simultaneously, and then boost the market demand of green products. The interest rate of the bank is not affected by this parameter. All the profits of the manufacturer, the platform, and the bank improve.

When the service commission fee increases, the manufacturer is forced to increase the unit selling price of green products and lower the level of green R&D. As the unit selling price increases, the market demand for the products decreases accordingly. The interest rate of the bank is not affected by the service commission fee. At this point, the increase in the manufacturer's sales revenue is less than the increase in the service commission fee, therefore its profit decreases. The increase in service commission fee enables the platform to invest more in DDM activities. When the service commission fee is below (above) a certain threshold, the increased revenue resulted from the increase in service commission fee exceeds (does not exceed) the increased investment of DDM investment, therefore the

platform's profit increases (decreases). It is further learned that the interest rate is not affected by the service commission fee, and the bank profit is only affected by the level of green R&D. Since the level of green R&D decreases with an increase in service commission fee, the profit of the bank also decreases.

*4.2. Reselling Model*

Under the reselling strategy, when the manufacturer finances from the bank, the bank first decides the financing interest rate, then the manufacturer decides the wholesale price of the product and the level of green R&D, and, finally, the platform decides the unit markup of the product and the level of DDM activities.

The profit functions of the manufacturer, the platform, and the bank are as follows:

$$\Pi_M^{r-bf}(w,e) = wD - \frac{e^2}{2} - \frac{\gamma e^2}{2} \tag{4}$$

$$\Pi_P^{r-bf}(m,t) = mD - \frac{t^2}{2} \tag{5}$$

$$\Pi_B^{a-bf}(\gamma) = \frac{\gamma e^2}{2} \tag{6}$$

**Proposition 2.** *The equilibrium decisions of reselling strategy financing from bank of platform supply chain with green R&D are:* $D^{r-bf*} = \frac{8-\alpha^2-4\beta^2}{4(2-\beta^2)(4-\alpha^2-2\beta^2)}$, $w^{r-bf*} = \frac{8-\alpha^2-4\beta^2}{4(4-\alpha^2-2\beta^2)}$, $m^{r-bf*} = \frac{8-\alpha^2-4\beta^2}{4(2-\beta^2)(4-\alpha^2-2\beta^2)}$, $p^{r-bf*} = \frac{(3-\beta^2)(8-\alpha^2-4\beta^2)}{4(2-\beta^2)(4-\alpha^2-2\beta^2)}$, $e^{r-bf*} = \frac{\alpha}{2(4-\alpha^2-2\beta^2)}$, $t^{r-bf*} = \frac{\beta(8-\alpha^2-4\beta^2)}{4(2-\beta^2)(4-\alpha^2-2\beta^2)}$, $\gamma^{r-bf*} = \frac{4-2\beta^2-\alpha^2}{4-2\beta^2}$, $\Pi_M^{r-bf*} = \frac{8-\alpha^2-4\beta^2}{8(2-\beta^2)(4-\alpha^2-2\beta^2)}$, $\Pi_P^{r-bf*} = \frac{(8-\alpha^2-4\beta^2)^2}{32(2-\beta^2)(4-\alpha^2-2\beta^2)^2}$, $\Pi_B^{r-bf*} = \frac{\alpha^2}{16(2-\beta^2)(4-\alpha^2-2\beta^2)}$.

The proof is given in Appendix B.

**Conclusion 2.** (*a*) $\frac{\partial D^{r-bf*}}{\partial \alpha} > 0$, $\frac{\partial w^{r-bf*}}{\partial \alpha} > 0$, $\frac{\partial m^{r-bf*}}{\partial \alpha} > 0$, $\frac{\partial p^{r-bf*}}{\partial \alpha} > 0$, $\frac{\partial e^{r-bf*}}{\partial \alpha} > 0$, $\frac{\partial t^{r-bf*}}{\partial \alpha} > 0$, $\frac{\partial \gamma^{r-bf*}}{\partial \alpha} < 0$, $\frac{\partial \Pi_M^{r-bf*}}{\partial \alpha} > 0$, $\frac{\partial \Pi_P^{r-bf*}}{\partial \alpha} > 0$, $\frac{\partial \Pi_B^{r-bf*}}{\partial \alpha} > 0$. (*b*) $\frac{\partial D^{r-bf*}}{\partial \beta} > 0$, $\frac{\partial w^{r-bf*}}{\partial \beta} > 0$, $\frac{\partial m^{r-bf*}}{\partial \beta} > 0$, $\frac{\partial p^{r-bf*}}{\partial \beta} > 0$, $\frac{\partial e^{r-bf*}}{\partial \beta} > 0$, $\frac{\partial t^{r-bf*}}{\partial \beta} > 0$, $\frac{\partial \gamma^{r-bf*}}{\partial \beta} < 0$, $\frac{\partial \Pi_M^{r-bf*}}{\partial \beta} > 0$, $\frac{\partial \Pi_P^{r-bf*}}{\partial \beta} > 0$, $\frac{\partial \Pi_B^{r-bf*}}{\partial \beta} > 0$.

Conclusion 2 shows that, in the case of the reselling strategy financing from the bank, both the consumers' sensitivities to the levels of green R&D and DDM activities have similar impacts on decision variables and profits. Specifically, as the consumers' sensitivity to the level of green R&D (the level of DDM activities) increases, and the manufacturer invests more to improve the green R&D level and then raises the wholesale price of green products. The platform is inspired to raise the level of DDM activities and unit markup (selling price) of the products. An increase in the levels of green R&D and DDM activities can make more potential consumers willing to buy green products. Then, both the manufacturer and the platform can benefit from an increase in these two parameters. Although the bank reduces the interest rate, its profit also improves due to an increase in the demand for green products.

## 5. Financing by Platform Model

*5.1. Agency Selling Model*

Under the agency selling strategy, when the manufacturer obtains funding from the platform, the platform decides the financing interest rate first, then the manufacturer decides the product selling price and the level of green R&D, and, finally, the platform decides the level of DDM activities.

The profit functions of manufacturer and platform are as follows:

$$\Pi_M^{a-pf}(p,e) = (p-u)D - \frac{e^2}{2} - \frac{\gamma e^2}{2} \tag{7}$$

$$\Pi_P^{a-pf}(t) = uD - \frac{t^2}{2} + \frac{\gamma e^2}{2} \tag{8}$$

**Proposition 3.** *To ensure that the financing interest rate is greater than zero, the following conditions should be satisfied:* $u < \frac{1}{3-\beta^2}$. *Then, the equilibrium decisions of agency selling strategy financing by platform are:* $D^{a-pf*} = \frac{4[1-u(1-\beta^2)]-\alpha^2[1-u(3-\beta^2)]}{4(2-\alpha^2)}$, $p^{a-pf*} = \frac{(4-\alpha^2)[1+u(1+\beta^2)]}{4(2-\alpha^2)}$,
$e^{a-pf*} = \frac{\alpha[1+u(1+\beta^2)]}{2(2-\alpha^2)}$, $t^{a-pf*} = \beta u$, $\gamma^{a-pf*} = \frac{(2-\alpha^2)[1-u(3-\beta^2)]}{2[1+u(1+\beta^2)]}$,
$\Pi_M^{a-pf*} = \frac{[1-u(1-\beta^2)]\{(4-\alpha^2)[1-u(1-\beta^2)]+2\alpha^2 u\}}{8(2-\alpha^2)}$,
$\Pi_P^{a-pf*} = \frac{\left[(\beta^2 u+1)^2 + 3u(3u-2) + 2\beta^2 u^2\right]\alpha^2 + 16u(1-u)}{16(2-\alpha^2)}$.

The proof is given in Appendix C.

**Conclusion 3.** (*a*) $\frac{\partial D^{a-pf*}}{\partial \alpha} > 0$, $\frac{\partial p^{a-pf*}}{\partial \alpha} > 0$, $\frac{\partial e^{a-pf*}}{\partial \alpha} > 0$, $\frac{\partial t^{a-pf*}}{\partial \alpha} = 0$, $\frac{\partial \gamma^{a-pf*}}{\partial \alpha} < 0$, $\frac{\partial \Pi_M^{a-pf*}}{\partial \alpha} > 0$, $\frac{\partial \Pi_P^{a-pf*}}{\partial \alpha} > 0$. (*b*) $\frac{\partial D^{a-pf*}}{\partial \beta} > 0$, $\frac{\partial p^{a-pf*}}{\partial \beta} > 0$, $\frac{\partial e^{a-pf*}}{\partial \beta} > 0$, $\frac{\partial t^{a-pf*}}{\partial \beta} > 0$, $\frac{\partial \gamma^{a-pf*}}{\partial \beta} > 0$, $\frac{\partial \Pi_M^{a-pf*}}{\partial \beta} > 0$, $\frac{\partial \Pi_P^{a-pf*}}{\partial \beta} > 0$. (*c*) $\frac{\partial p^{a-pf*}}{\partial u} > 0$, $\frac{\partial e^{a-pf*}}{\partial u} > 0$, $\frac{\partial t^{a-pf*}}{\partial u} > 0$, $\frac{\partial \gamma^{a-pf*}}{\partial u} < 0$; $\frac{\partial D^{a-pf*}}{\partial u} > 0$ *when* $\beta^2 \geq \frac{1}{3}$ *and* $\alpha^2 > \frac{4(1-\beta^2)}{3-\beta^2}$, *otherwise* $\frac{\partial D^{a-pf*}}{\partial u} < 0$; $\frac{\partial \Pi_M^{a-pf*}}{\partial u} < 0$ *when* $\alpha^2 < \frac{4-4\beta^2(2-\beta^2)}{3-2\beta^2-\beta^2(2-\beta^2)}$, *otherwise* $\frac{\partial \Pi_M^{a-pf*}}{\partial u} > 0$; $\frac{\partial \Pi_P^{a-bf*}}{\partial u} \geq 0$ *when* $u < \frac{1}{3-\beta^2}$.

Conclusion 3 shows that, in the case of agent selling strategy financing from the platform, when the consumers' sensitivity to the level of green R&D increases, the platform does not change the level of DDM activities but reduces the interest rate to encourage the manufacturer to invest more in improving the green R&D level; accordingly, the manufacturer raises the green R&D level and then increases the price of green products. The enhancement of the green R&D level helps to increase the market demand for green products. Both parties can obtain more profits. When the consumers' sensitivity to the level of DDM activities increases, the platform increases both the level of DDM activities and interest rate, and the manufacturer also spends more in green R&D and raises the price. An increase in green R&D and DDM levels attracts more consumers to purchase green products. Therefore, the platform can gain more profits from the increase in service commission fee and loan interests, while the manufacturer benefits from the increase in the market demand and selling prices for green products.

Similar to the agency selling model financing by bank, when the unit service commission fee provided by manufacturer to the platform increases, both the unit selling price of the green products of the manufacturer and DDM level of the platform increase. However, different from the case financing by bank, the level of green R&D increases and the interest rate decreases in unit service commission fee under the case financing by the platform. The reason for this is that, when the platform provides loan services to the manufacturer, it can make profits from both the service commission fee and the loan interest. Thus, as the unit commission fee increases, it gives up a certain loan interests to encourage the manufacturer to improve the level of R&D and yield more market demand. The simultaneous increase in unit commission fee and market demand will be beneficial to the platform. In contrast, when the bank provides loans to the manufacturer, since the loan interest of the bank is independent of the unit commission fee, the manufacturer's motivation to improve the level of R&D reduces as the commission fee increases. Only when both the consumers'

sensitivities to DDM activities and green R&D level is relatively higher, the market demand increases with the increase in unit commission fee, otherwise the demand decreases in unit commission fee. The impact of unit commission fee on the profit of the manufacturer depends on the consumers' sensitivity to green R&D. Specifically, when the consumers' sensitivity to green R&D is lower (higher) than a certain threshold, the profit of the manufacturer decreases (increases) in the consumers' sensitivity to green R&D. Moreover, when the unit service commission fee does not exceed a threshold which is co-determined by both the consumers' sensitivities to green R&D and DDM activities, the profit of the platform increases in the unit commission fee. This result reveals that a unit commission fee that is too high is detrimental to the platform because it is forced to reduce the interest rate below zero. By contrast, a moderate unit commission fee will benefit the platform.

**Conclusion 4.** $\Pi_M^{a-bf*} < \Pi_M^{a-pf*}$, $\Pi_P^{a-bf*} < \Pi_P^{a-pf*}$, $e^{a-bf*} < e^{a-pf*}$ *when* $u < \frac{1}{3-\beta^2}$.

Conclusion 4 shows that, under the agent selling strategy, if unit service commission fee is less than the threshold $\frac{1}{3-\beta^2}$, the manufacturer will prefer to obtain financing from the platform rather than the bank, and this financing strategy will also realize the profit improvement of the platform. The reason for this is that, if the unit service commission fee is relatively low, when the manufacturer obtains financing from the platform, the platform can use the loan proceeds to invest in DDM activities, which boosts the market demand for green products. Then, the manufacturer is motivated to improve the level of green R&D and further enhance the market demand. Thus, compared to the case of financing by bank, the manufacturer is willing to finance from the platform. However, when the unit service commission fee exceeds the threshold, the manufacturer needs to pay more service commission fee to the platform; at this time, the platform has to reduce the interest to zero to avoid driving the manufacturer out of the supply chain operations. Hence, for the platform, there are more kicks than halfpence when the unit service commission fee is relatively higher.

### 5.2. Reselling Model

Under the reselling strategy, when the manufacturer obtains funding from the platform, the platform first decides the financing interest rate, then the manufacturer decides the wholesale price of the product and the level of green R&D, and, finally, the platform decides the unit markup of the product and the level of DDM activities.

The profit functions of manufacturer and platform are as follows:

$$\Pi_M^{r-pf}(w,e) = wD - \frac{e^2}{2} - \frac{\gamma e^2}{2} \tag{9}$$

$$\Pi_P^{r-pf}(m,t) = mD - \frac{t^2}{2} + \frac{\gamma e^2}{2} \tag{10}$$

**Proposition 4.** *The equilibrium decisions of reselling strategy financing by the platform of platform supply chain under green R&D is:* $D^{r-pf*} = \frac{24-5\alpha^2-12\beta^2}{6(2-\beta^2)(8-3\alpha^2-4\beta^2)}$, $w^{r-pf*} = \frac{24-5\alpha^2-12\beta^2}{6(8-3\alpha^2-4\beta^2)}$, $m^{r-pf*} = \frac{24-5\alpha^2-12\beta^2}{6(2-\beta^2)(8-3\alpha^2-4\beta^2)}$, $p^{r-pf*} = \frac{(3-\beta^2)(24-5\alpha^2-12\beta^2)}{6(2-\beta^2)(8-3\alpha^2-4\beta^2)}$, $e^{r-pf*} = \frac{4\alpha}{12(2-\beta^2)-9\alpha^2}$, $t^{r-pf*} = \frac{\beta(24-5\alpha^2-12\beta^2)}{6(2-\beta^2)(8-3\alpha^2-4\beta^2)}$, $\gamma^{r-pf*} = \frac{8-5\alpha^2-4\beta^2}{8(2-\beta^2)}$, $\Pi_M^{r-pf*} = \frac{24-5\alpha^2-12\beta^2}{12(2-\beta^2)(8-3\alpha^2-4\beta^2)}$, $\Pi_P^{r-pf*} = \frac{36(2-\beta^2)+5\alpha^2}{72(2-\beta^2)(8-3\alpha^2-4\beta^2)}$.

The proof is given in Appendix D.

**Conclusion 5.** *(a)* $\frac{\partial D^{r-pf*}}{\partial \alpha} > 0$, $\frac{\partial w^{r-pf*}}{\partial \alpha} > 0$, $\frac{\partial m^{r-pf*}}{\partial \alpha} > 0$, $\frac{\partial p^{r-pf*}}{\partial \alpha} > 0$, $\frac{\partial e^{r-pf*}}{\partial \alpha} > 0$, $\frac{\partial t^{r-pf*}}{\partial \alpha} > 0$, $\frac{\partial \gamma^{r-pf*}}{\partial \alpha} < 0$, $\frac{\partial \Pi_M^{r-pf}}{\partial \alpha} > 0$, $\frac{\partial \Pi_P^{r-pf}}{\partial \alpha} > 0$. *(b)* $\frac{\partial D^{r-pf*}}{\partial \beta} > 0$, $\frac{\partial w^{r-pf*}}{\partial \beta} > 0$, $\frac{\partial m^{r-pf*}}{\partial \beta} > 0$, $\frac{\partial p^{r-pf*}}{\partial \beta} > 0$, $\frac{\partial e^{r-pf*}}{\partial \beta} > 0$, $\frac{\partial t^{r-pf*}}{\partial \beta} > 0$, $\frac{\partial \gamma^{r-pf*}}{\partial \beta} < 0$, $\frac{\partial \Pi_M^{r-pf}}{\partial \beta} > 0$, $\frac{\partial \Pi_P^{r-pf}}{\partial \beta} > 0$.

Conclusion 5 shows that, in the case of the reselling strategy financing by the platform, the two parameters of the consumers' sensitivities to the levels of green R&D and DDM activities have the same impacts on decision variables and profits as those in the case of the reselling model financing by the bank. We will not repeat it again.

**Conclusion 6.** $\Pi_M^{r-bf*} < \Pi_M^{r-pf*}$, $\Pi_P^{r-bf*} < \Pi_P^{r-pf*}$, $e^{r-bf*} < e^{r-pf*}$.

Conclusion 6 shows that, under the reselling strategy, the manufacturer will always prefer to obtain financing from the platform rather than the bank regardless of the unit service commission fee. Moreover, under this selling mode, the profit of the platform and green R&D level with a platform financing channel are also higher than those with a bank financing channel.

## 6. Numerical Analysis

The following is a numerical analysis to verify the above propositions, and then explore the management implications of financing channels and selling modes of the platform supply chain.

### 6.1. Sensitivity Analysis

From Figures 2–4, we can find that, in both financing channels, when the unit service commission fee is relatively low, the manufacturer tends to choose the agency selling strategy, since, under this strategy, the manufacturer can directly sell green products at a lower price and set a higher green R&D level to further boost the market demand compared to the reselling strategy. In contrast, when the unit service commission fee is relatively high, the profit of the manufacturer under this strategy is lower than that under the reselling strategy due to the high service commission fee, therefore the manufacturer chooses the reselling strategy. For the platform, when the unit service commission fee is relatively low, it cannot obtain enough of a commission fee, therefore its profit under agency selling strategy is lower than that under the reselling strategy. With an increase in the unit service commission fee, the profit of the platform under the agency selling strategy will gradually become close to and surpass that of the reselling strategy. However, a unit service commission fee that is not higher is better for the platform. Specifically, when unit service commission fee exceeds a certain threshold, because the manufacturer prices the green products too high and sets the green R&D level too low, the market demand of green products drops dramatically, and then the platform also suffer a loss. Hence, the platform will turn back to the reselling strategy.

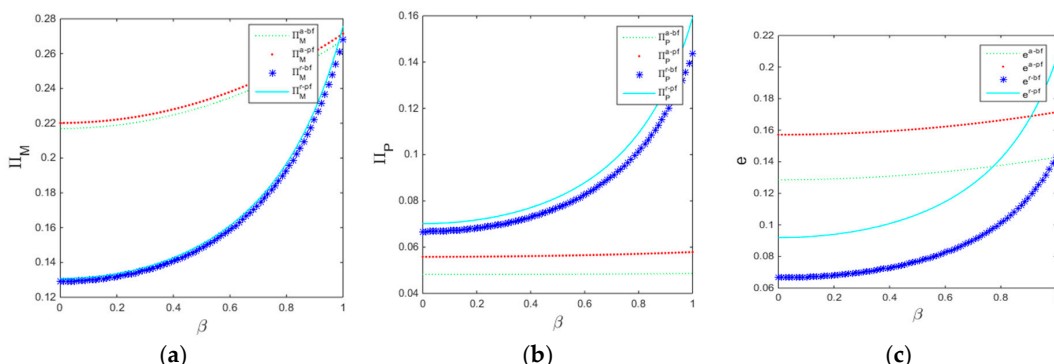

**Figure 2.** Product green R&D level and profits of manufacturer and platform when $u = 0.1$, $\alpha = 0.5$: (**a**) Manufacturer's profit. (**b**) Platform's profit. (**c**) Product green R&D level.

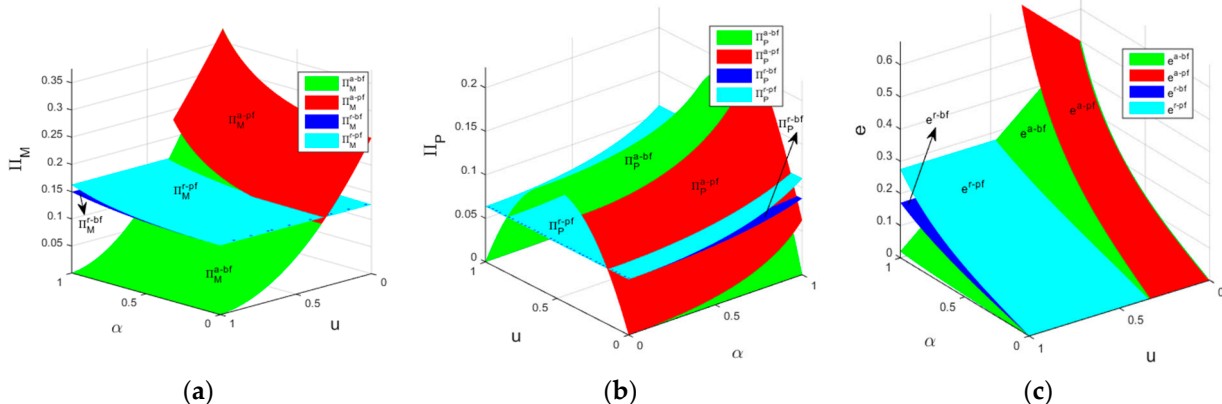

**Figure 3.** Product green R&D level and profits of manufacturer and platform when $\beta = 0.2$: (**a**) Manufacturer's profit. (**b**) Platform's profit. (**c**) Product green R&D level.

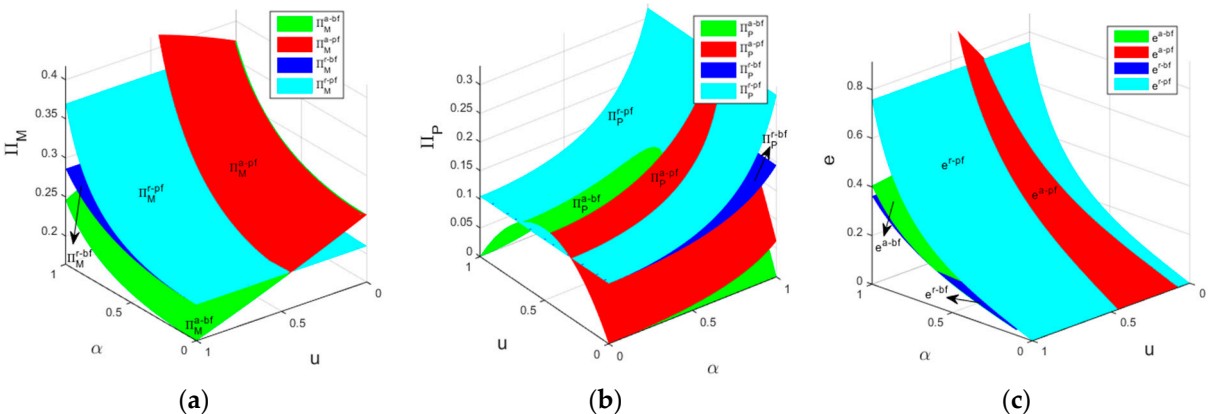

**Figure 4.** Product green R&D level and profits of manufacturer and platform when $\beta = 0.9$: (**a**) Manufacturer's profit. (**b**) Platform's profit. (**c**) Product green R&D level.

As shown in Figures 3 and 4, under the agent selling mode, if the unit service commission fee is in a reasonable region, the manufacturer prefers to finance from the platform, and this financing strategy is also in favor of the platform. Under the reselling mode, the manufacturer financing from the platform is superior to financing from the platform for both parties no matter the value of the unit service commission fee. This is consistent with Conclusions 4 and 6.

### 6.2. Financing Channel and Selling Mode Choice

Figures 5 and 6 show that the impacts of the sensitivity coefficient of DDM activities on the choices of the selling mode and financing channel. We can find that, no matter the value of the sensitivity coefficient of DDM activities, (1) when the service commission fee is relatively small, the platform prefers to finance the manufacturer and adopts a reselling mode, and the manufacturer also hopes to obtain funds from the platform but tends to agency selling mode. This result can be partly obtained from Conclusions 4 and 6, namely that, with a small sensitivity coefficient of DDM activities, both the manufacturer and the platform tend towards the platform financing channel. As for the selling mode, under the agency selling mode, the manufacturer pays a small amount of service commission fee to the platform and can directly sell more products at a lower price without a double marginalization effect. Thus, agency selling mode is preferable to the manufacturer than the reselling mode. From the perspective of the platform, it is more inclined to choose reselling mode because it can only earn a little service commission fee by the agency selling mode. Given that the selling mode is mainly determined by the platform, the equilibrium

selling mode and financing channel in this case is the reselling mode and platform financing channel; (2) when the service commission fee is medium ($< \frac{1}{3-\beta^2}$), both the manufacturer and the platform tend to choose the agency selling mode and platform financing channel. This is because, on one hand, a medium level of unit service commission fee increases the profit of the platform; on the other hand, the improvement of the level of DDM activities by the platform boosts the market demand, therefore the manufacturer is also better off; (3) when the unit service commission fee is very high, both the manufacturer and the platform tend to choose the reselling mode and platform financing channel. This is because, under the agency selling mode, a very high level of unit service commission fee forces the manufacturer to raise the selling price and reduce the green R&D level by a wide margin, and then leads to a sharp reduction in the sales volume of green products. Thus, both parties are worse off in this circumstance and prefer the reselling mode and platform financing channel.

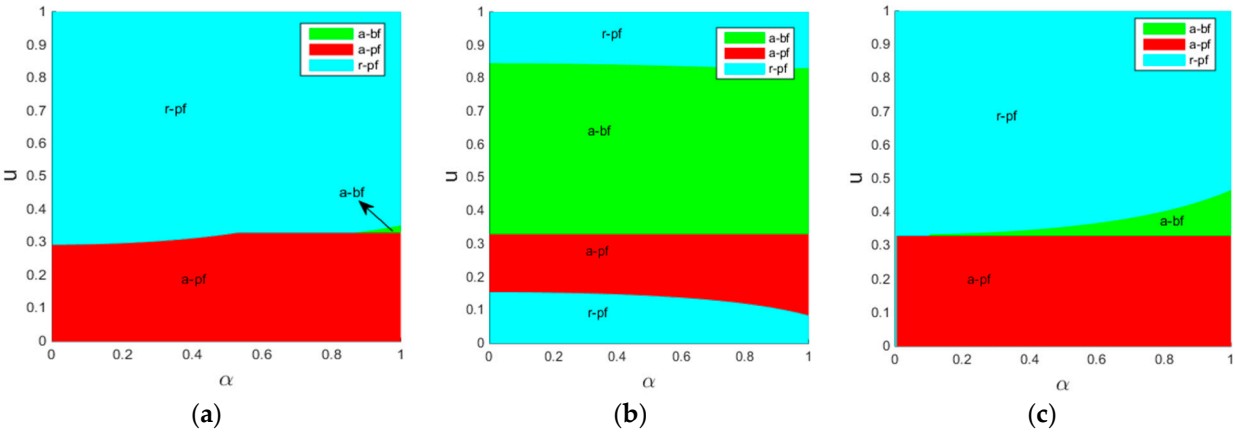

**Figure 5.** Optimal green level and manufacturer and platform strategy selection when $\beta = 0.2$: (**a**) Manufacturer. (**b**) Platform. (**c**) Product green R&D level.

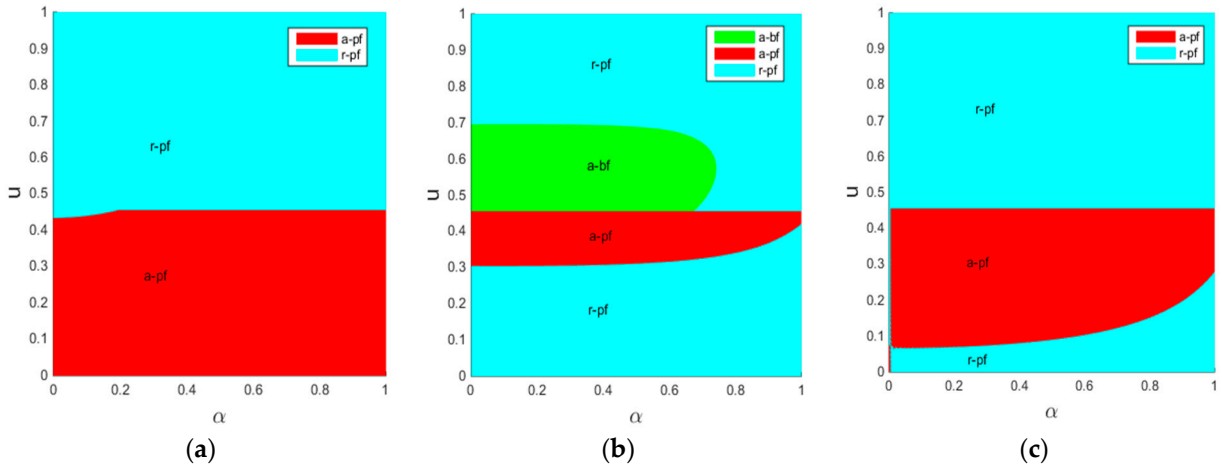

**Figure 6.** Optimal green level and manufacturer and platform strategy selection when $\beta = 0.9$: (**a**) Manufacturer. (**b**) Platform. (**c**) Product green R&D level.

The choices of the selling mode and financing channel in the following scenarios depend on the sensitivity coefficient of DDM activities. Specifically, when the sensitivity coefficient of DDM activities is small, if the unit service commission fee is above medium ($> \frac{1}{3-\beta^2}$), in most conditions, the manufacturer tends to choose reselling mode and platform financing channel, while the platform prefers the agency selling mode and bank financing channel. Thus, it is difficult for both parties to reach an agreement on the selling mode and

financing channel. The reason for this lies in the fact that, for the manufacturer, in normal circumstances, an above-medium unit service commission fee under an agency selling mode makes him turn to a reselling mode; besides that based on Conclusions 4 and 6, the platform financing channel is strictly superior to bank financing channel for the manufacturer under this selling mode. For the platform, an above-medium unit service commission fee makes itself benefit from an agency selling mode, and the fact that the consumers are less sensitive to DDM activities reduces its motivation to develop DDM technologies, therefore the platform needs not rely on loan interests to reach a high level of DDM activities. There is an exception that, when the consumers' sensitivity to green R&D is extremely high, both parties prefer to choose the agency selling mode and bank financing channel. This is attributed to the fact that, when the sensitivity coefficient of DDM activities is small, the advantage of the market demand under the agency selling mode to that under the reselling mode enlarges with an increase in the consumers' sensitivity to the level of green R&D ($\partial D^{a-bf*}/\partial\alpha > D^{r-pf*}/\partial\alpha$). Hence, when this parameter is sufficiently large, the manufacturer will still prefer to choose the agency selling mode. In addition, since the unit service commission fee is higher than $\frac{1}{3-\beta^2}$, the manufacturer hopes to receive finance from the bank in this selling mode.

When the sensitivity coefficient of DDM activities is large, if the unit service commission fee is above medium and the consumers' sensitivity to green R&D is small, the manufacturer tends to choose the reselling mode and platform financing channel, while the platform prefers the reselling mode and bank financing channel. The reason for this is that, as mentioned before, an above-medium level of unit service commission fee under an agent selling mode and platform financing channel makes both parties worse off. Then, the manufacturer turns to the reselling mode. In the reselling mode, when the manufacturer finances from the platform, the platform can utilize the loan interests to invest in DDM technologies and promote the market demand markedly in the case that the sensitivity coefficient of DDM activities is high. A substantial increase in the demand is beneficial to the manufacture in turn. However, from another angle, because the consumers' sensitivity to green R&D is small, an increase in the demand is due mainly to the investment of the platform, leading to a profit increment in the platform brought on by the fact that an increased demand cannot offset the increased investment of DDM activities. Therefore, the platform requires the manufacturer to finance from the bank. In addition, since the unit service commission fee is above medium, it is willing to adopt an agency selling mode. On the contrary, if both the consumers' sensitivities to DDM activities and green R&D are large, the efficiencies of investment of both parties are high under the reselling mode and platform financing channel; then, this strategy is the best for both parties.

## 7. Conclusions and Future Study

With the deterioration of the global environment, more and more enterprises are engaged in the production and innovation of green products. Moreover, in the platform supply chain, many scholars have discovered the advantages of the platform economy and have studied how the DDM activities of the platform positively affect the operations of the supply chain [10]. Therefore, we consider how the platform can give play to its advantages to improve the sales of green products in line with the current research trend. In addition, the rapid development of the platform also brings on new financing methods for manufacturers [37]. In view of the lack of comprehensive research on green R&D, the selling mode, and the manufacturer financing channels in an online marketplace, this paper establishes four platform supply chain decision models under two financing channels and two selling modes, and then compares and analyzes the equilibrium results of different models. By analyzing the impacts of DDM and green R&D on the selling mode and financing channel from the perspectives of both the manufacturer and the platform, this paper fills in the research gaps in the existing literature. The main conclusions are as follows:

The service commission fee that is charged by the platform in the agent selling mode, consumers' sensitivities to green R&D and DDM activities are the main factors influ-encing

the product green R&D level and the choices of sales modes and financing channels by the manufacturer and the platform. Specifically, the increase in both consumers' sensitivities to green R&D and DDM activities can increase the level of green R&D and significantly improve the profits of the manufacturer and the platform. Under agency selling mode, if the unit service commission fee is low, the platform financing channel can motivate the platform to invest more in DDM activities and then enhance the market demand for green products. In turn, the manufacturer is encouraged to improve the level of green R&D. Both parties prefer platform financing rather than bank financing; on the contrary, when the unit service commission fee is high, platform financing hurts both parties' interests and bank financing is better. Under the reselling strategy, compared with bank financing, the manufacturer is willing to set a higher level of green R&D and both parties can obtain more profits in the case of platform financing. In reality, the Amazon platform, in conjunction with Lendistry, a commercial lender, is offering a new type of business financing pattern that can help companies grow. In addition, since the green products generally have relatively low service commission fees, Amazon prefers to adopt the reselling mode in most cases. This phenomenon is consistent with the results of our analysis (illustrated in blue in Figures 5 and 6). The above conclusions are different from those in Zhen et al. [37]. They consider the case that a capital-constrained manufacturer produces ordinary products and the platform does not engage in DDM activities, and find that platform financing is always better than bank financing for the manufacturer.

There are four situations in which the manufacturer and the platform can agree on a same strategy and on the same selling mode and financing channel, and the level of green R&D is the optimal. In detail, when the service commission fee is medium, both parties tend to choose agency selling and platform financing; when the unit service commission fee is high, both parties prefer reselling and platform financing; when the sensitivity coefficient of DDM activities is small, the unit service commission fee is above medium, and consumers are highly sensitive to green R&D, both parties are more inclined to choose agency selling and bank financing. If consumers are sensitive to DDM activities and green R&D, and the unit service commission fee is above medium, the investment efficiency of both parties is the highest under reselling and platform financing, and this strategy is the best for both parties.

We would like to further explore the following directions. First of all, it is assumed that, in this paper, that the platform provides only one selling mode; however, in reality, the platform can provide two modes of agency selling and reselling simultaneously. Therefore, future research can consider the case that the two selling modes co-exist. Secondly, this paper merely considers the green R&D of the manufacturer to produce one green product. However, in practical operations, green products can be divided into production-intensive products and R&D-intensive products. Thus, the innovation of different green product types can be studied in the future. In addition, government policies, as a common and effective method to guide the manufacturer to better implement green production and R&D, can also be incorporated into our model in future studies.

**Author Contributions:** Conceptualization, Z.M. and Y.X.; methodology, Z.M.; software, Y.X. and K.L.; validation, Y.X. and H.S.; formal analysis, Z.M.; investigation, K.L.; resources, H.S. and Q.G.; data curation, Z.M. and Y.X.; writing—original draft preparation, Y.X.; writing—review and editing, Y.X.; visualization, Z.M.; supervision, H.S.; project administration, H.S.; funding acquisition, Z.M. All authors have read and agreed to the published version of the manuscript.

**Funding:** This work was supported in part by the Nature Science Foundation of Shandong Province under Grant ZR2017MG015 and ZR2017BG002.

**Institutional Review Board Statement:** Not applicable.

**Informed Consent Statement:** Not applicable.

**Data Availability Statement:** Not applicable.

**Conflicts of Interest:** The authors declare no conflict of interest.

## Appendix A

**Proof of Proposition 1.** By the standard backward induction, we can first verify that $\frac{\partial^2 \Pi_P^{a-bf}(t)}{\partial t^2} = -1 < 0$; thus, the profit of the platform is a concave function with $t$. By solving $\frac{\partial \Pi_P^{a-bf}(t)}{\partial t} = 0$, we obtain the platform's response decision $t^{a-bf} = \beta u$. By substituting the $t^{a-bf} = \beta u$ into the profit function of the manufacturer $\Pi_M^{a-bf}(p,e)$ and taking the derivative with respect to $p$ and $e$, we can see that the Hessian matrix of $\Pi_M^{a-bf}(p,e)$ is $\begin{pmatrix} -2 & \alpha \\ \alpha & -(1+\gamma) \end{pmatrix}$. It is easy to see that the Hessian matrix is negative-definite when $2(1+\gamma) - \alpha^2 > 0$. Then, by solving $\frac{\partial \Pi_M^{a-bf}(p,e)}{\partial p} = 0$ and $\frac{\partial \Pi_M^{a-bf}(p,e)}{\partial e} = 0$ simultaneously, we can obtain $e^{a-bf*} = \frac{\alpha[1-u(1-\beta^2)]}{2(1+\gamma)-\alpha^2}$ and $p^{a-bf*} = \frac{(1+\gamma)[1+u(1+\beta^2)]-\alpha^2 u}{2(1+\gamma)-\alpha^2}$. By substituting $e^{a-bf*}$ and $p^{a-bf*}$ into $t^{a-bf}$, and then substituting all of them into the profit functions, we can obtain $\Pi_M^{a-bf*} = \frac{(1+\gamma)[1-u(1-\beta^2)]^2}{2[2(1+\gamma)-\alpha^2]}$, $\Pi_P^{a-bf*} = \frac{u[\alpha^2\beta^2 u + 2(1+\gamma)(1-u)]}{2[2(\gamma+1)-\alpha^2]}$, $\Pi_B^{a-bf*} = \frac{\gamma\alpha^2[1-u(1-\beta^2)]^2}{2[2(\gamma+1)-\alpha^2]^2}$.

For the bank, since $\frac{\partial^2 \Pi_B^{a-bf}(\gamma)}{\partial \gamma^2} = -\frac{(2-\gamma-\alpha^2)\{2\alpha[1-u(1-\beta^2)]\}^2}{[2(1+\gamma)-\alpha^2]^4} < 0$, its profit is a concave function of $\gamma$. Let $\frac{\partial \Pi_B^{a-bf}(\gamma)}{\partial \gamma} = 0$, we can obtain $\gamma^{a-bf*} = \frac{2-\alpha^2}{2}$. Substituting the optimal interest rate $\gamma^{a-bf*} = \frac{2-\alpha^2}{2}$ into the response functions yields $p^{a-bf*} = \frac{(4-\alpha^2)[1+u(1+\beta^2)]-2\alpha^2 u}{4(2-\alpha^2)}$, $e^{a-bf*} = \frac{\alpha[1-u(1-\beta^2)]}{2(2-\alpha^2)}$, and $t^{a-bf*} = \beta u$. Finally, by substituting $p^{a-bf*}$, $e^{a-bf*}$, $t^{a-bf*}$, and $\gamma^{a-bf*}$ into the market demand and the profits of the manufacturer, the platform, and the bank, Proposition 1 can be obtained. $\square$

## Appendix B

**Proof of Proposition 2.** Through the standard backward induction, we first derive that the Hessian matrix of $\Pi_P^{r-bf}(m,t)$ is $\begin{pmatrix} -1 & \beta \\ \beta & -2 \end{pmatrix}$. It is easy to verify that the Hessian matrix is negative-definite when $2 - \beta^2 > 0$. Let $\frac{\partial \Pi_P^{r-bf}(m,t)}{\partial m} = 0$ and $\frac{\partial \Pi_P^{r-bf}(m,t)}{\partial t} = 0$, we can obtain $m^{r-bf} = \frac{1-w+\alpha e}{2-\beta^2}$ and $t^{r-bf} = \frac{\beta(1-w+\alpha e)}{2-\beta^2}$. By substituting $m^{r-bf}$ and $t^{r-bf}$ into the profit function of the manufacturer $\Pi_M^{r-bf}(w,e)$ and taking the derivative with respect to $w$ and $e$, we can verify that the Hessian matrix of $\Pi_M^{r-bf}(w,e)$ is $\begin{pmatrix} -2 & \alpha \\ \alpha & -(1+\gamma) \end{pmatrix}$. Thus, the Hessian matrix is negative-definite when $2(1+\gamma) - \alpha^2 > 0$. Let $\frac{\partial \Pi_M^{r-bf}(w,e)}{\partial w} = 0$ and $\frac{\partial \Pi_M^{r-bf}(w,e)}{\partial e} = 0$, we can obtain $w^{r-bf*} = \frac{(2-\beta^2)(\gamma+1)}{2(2-\beta^2)(\gamma+1)-\alpha^2}$ and $e^{r-bf*} = \frac{\alpha}{2(2-\beta^2)(\gamma+1)-\alpha^2}$. Substituting $w^{r-bf*}$ and $e^{r-bf*}$ into $m^{r-bf}$ and $t^{r-bf}$ yields $m^{r-bf*} = \frac{(\gamma+1)}{2(2-\beta^2)(\gamma+1)-\alpha^2}$ and $t^{r-bf*} = \frac{\beta(\gamma+1)}{2(2-\beta^2)(\gamma+1)-\alpha^2}$. Then, from $w^{r-bf*}$, $m^{r-bf*}$, $e^{r-bf*}$, and $t^{r-bf*}$, we can obtain $\Pi_M^{r-bf*} = \frac{(1+\gamma)}{2[2(2-\beta^2)(1+\gamma)-\alpha^2]}$, $\Pi_P^{r-bf*} = \frac{(2-\beta^2)(1+\gamma)^2}{2[\alpha^2-2(2-\beta^2)(1+\gamma)]^2}$, $\Pi_B^{r-bf*} = \frac{\gamma\alpha^2}{2[\alpha^2-2(2-\beta^2)(\gamma+1)]^2}$.

For the bank, since $\frac{\partial^2 \Pi_B^{r-bf}(\gamma)}{\partial^2 \gamma} = \frac{4\alpha^2(2-\beta^2)[\alpha^2-(2-\gamma)(2-\beta^2)]}{[\alpha^2-2(2-\beta^2)(\gamma+1)]^4} < 0$, the bank's profit is a concave function of $\gamma$. Let $\frac{\partial \Pi_B^{a-bf}(\gamma)}{\partial \gamma} = 0$, we can obtain $\gamma^{r-bf*} = \frac{4-\alpha^2-2\beta^2}{2(2-\beta^2)}$. Substituting the optimal interest rate $\gamma^{r-bf*} = \frac{4-\alpha^2-2\beta^2}{2(2-\beta^2)}$ into the response functions yields

$w^{r-bf*} = \frac{8-\alpha^2-4\beta^2}{4(4-\alpha^2-2\beta^2)}$, $m^{r-bf*} = \frac{8-\alpha^2-4\beta^2}{4(2-\beta^2)(4-\alpha^2-2\beta^2)}$, $p^{r-bf*} = \frac{(3-\beta^2)(8-\alpha^2-4\beta^2)}{4(2-\beta^2)(4-\alpha^2-2\beta^2)}$, $e^{r-bf*} = \frac{\alpha}{2(4-\alpha^2-2\beta^2)}$, and $t^{r-bf*} = \frac{\beta(8-\alpha^2-4\beta^2)}{4(2-\beta^2)(4-\alpha^2-2\beta^2)}$. Finally, by substituting $w^{r-bf*}$, $m^{r-bf*}$, $e^{r-bf*}$, $t^{r-bf*}$, and $\gamma^{r-bf*}$ into the market demand and the profits of the manufacturer, the platform, and the bank, Proposition 2 can be obtained. □

**Appendix C**

**Proof of Proposition 3.** Through the standard backward induction, we first verify that $\frac{\partial^2 \Pi_P^{a-pf}(t)}{\partial t^2} = -1 < 0$; thus, the profit of the platform is a concave function with $t$. By solving $\frac{\partial \Pi_P^{a-pf}(t)}{\partial t} = 0$ we can obtain the platform's response decision $t^{a-pf} = \beta u$. By substituting $t^{a-pf} = \beta u$ into the profit function of the manufacturer $\Pi_M^{a-pf}(p,e)$ and taking the derivative with respect to $p$ and $e$, we can derive that the Hessian matrix of $\Pi_M^{a-pf}(p,e)$ is $\begin{pmatrix} -2 & \alpha \\ \alpha & -(1+\gamma) \end{pmatrix}$. It is easy to see that the Hessian matrix is negative-definite when $2(1+\gamma) - \alpha^2 > 0$. Then, let $\frac{\partial \Pi_M^{a-pf}(p,e)}{\partial p} = 0$ and $\frac{\partial \Pi_M^{a-pf}(p,e)}{\partial e} = 0$, and we obtain $e^{a-pf*} = \frac{\alpha[1-u(1-\beta^2)]}{2(1+\gamma)-\alpha^2}$ and $p^{a-pf*} = \frac{(1+\gamma)[1+u(1+\beta^2)]-\alpha^2 u}{2(1+\gamma)-\alpha^2}$. By substituting $e^{a-pf*}$ and $p^{a-pf*}$ into $t^{a-pf*}$ and then substituting all of them into the profit functions, we can obtain $\Pi_M^{a-pf*} = \frac{(\gamma+1)[1-u(1-\beta^2)]^2}{2[2(\gamma+1)-\alpha^2]}$, $\Pi_P^{a-pf*} = \frac{\alpha^2\beta^2 u^2(2-\alpha^2)+2u(1-u)[2(\gamma+1)^2-\alpha^2]+\gamma\alpha^2[u^2(3+\beta^4)-2u(2-\beta^2)+1]}{2[2(\gamma+1)-\alpha^2]}$.

For the platform, $\frac{\partial^2 \Pi_P^{a-pf}(\gamma)}{\partial \gamma^2} = \frac{4\alpha^2[1-u(1-\beta^2)]\{(2-\alpha^2)[u(2-\beta^2)-1]+\gamma[u(\beta^2+1)+1]\}}{[2(\gamma+1)-\alpha^2]^4}$, therefore its profit is a concave function of $\gamma$ when $\gamma < \frac{(2-\alpha^2)[1-u(2-\beta^2)]}{1+u(1+\beta^2)}$. Let $\frac{\partial \Pi_P^{a-bf}(\gamma)}{\partial \gamma} = 0$, we can obtain $\gamma^{a-pf*} = \frac{(2-\alpha^2)[1-u(3-\beta^2)]}{2[1+u(1+\beta^2)]}$. In order to ensure that the financing interest rate is greater than 0, $u < \frac{1}{3-\beta^2}$ should be satisfied. Substituting the optimal interest rate into the response functions yields $p^{a-pf*} = \frac{(4-\alpha^2)[1+u(1+\beta^2)]}{4(2-\alpha^2)}$, $e^{a-pf*} = \frac{\alpha[1+u(1+\beta^2)]}{2(2-\alpha^2)}$, and $t^{a-pf*} = \beta u$. Finally, by substituting $p^{a-pf*}$, $e^{a-pf*}$, $t^{a-pf*}$, and $\gamma^{a-pf*}$ into the market demand and the profits of the manufacturer and the platform, Proposition 3 can be obtained. □

**Appendix D**

**Proof of Proposition 4.** By the standard backward induction, we first derive that the Hessian matrix of $\Pi_P^{r-pf}(m,t)$ is $\begin{pmatrix} -1 & \beta \\ \beta & -2 \end{pmatrix}$. It is easy to find that the Hessian matrix is negative-definite when $2 - \beta^2 > 0$. Let $\frac{\partial \Pi_P^{r-pf}(m,t)}{\partial m} = 0$ and $\frac{\partial \Pi_P^{r-pf}(m,t)}{\partial t} = 0$, we can obtain $m^{r-pf} = \frac{1-w+\alpha e}{2-\beta^2}$ and $t^{r-pf} = \frac{\beta(1-w+\alpha e)}{2-\beta^2}$. By substituting $m^{r-pf}$ and $t^{r-pf}$ into the profit function of the manufacturer $\Pi_M^{r-pf}(w,e)$ and taking the derivative with respect to $w$ and $e$, we can derive that the Hessian matrix of $\Pi_M^{r-pf}(w,e)$ is $\begin{pmatrix} -2 & \alpha \\ \alpha & -(1+\gamma) \end{pmatrix}$. The Hessian matrix is negative-definite when $2(1+\gamma) - \alpha^2 > 0$. Let $\frac{\partial \Pi_M^{r-pf}(w,e)}{\partial w} = 0$ and $\frac{\partial \Pi_M^{r-pf}(w,e)}{\partial e} = 0$, we obtain $w^{r-pf*} = \frac{(2-\beta^2)(\gamma+1)}{2(2-\beta^2)(\gamma+1)-\alpha^2}$ and $e^{r-pf*} = \frac{\alpha}{2(2-\beta^2)(\gamma+1)-\alpha^2}$.



Substituting $w^{r-pf*}$ and $e^{r-pf*}$ into $m^{r-pf}$ and $t^{r-pf}$ yields $m^{r-pf*} = \frac{(\gamma+1)}{2(2-\beta^2)(\gamma+1)-\alpha^2}$ and $t^{r-pf*} = \frac{\beta(\gamma+1)}{2(2-\beta^2)(\gamma+1)-\alpha^2}$. From $w^{r-pf*}$, $m^{r-pf*}$, $e^{r-pf*}$, and $t^{r-pf*}$, we can obtain $\Pi_M^{r-pf*} = \frac{(1+\gamma)}{2[2(2-\beta^2)(1+\gamma)-\alpha^2]}$, $\Pi_P^{r-pf*} = \frac{(2-\beta^2)(1+\gamma^2)+(4+\alpha^2-2\beta^2)\gamma}{2[\alpha^2-2(2-\beta^2)(\gamma+1)]^2}$.

For the platform, $\frac{\partial^2 \Pi_P^{r-pf}(\gamma)}{\partial \gamma^2} = \frac{\alpha^2(2-\beta^2)[4(2-\beta^2)(2\gamma-1)+5\alpha^2]}{[2(2-\beta^2)(1+\gamma)-\alpha^2]^4}$; therefore, its profit is a concave function of $\gamma$ when $\gamma < \frac{4(2-\beta^2)-5\alpha^2}{8(2-\beta^2)}$. Let $\frac{\partial \Pi_B^{r-pf}(\gamma)}{\partial \gamma} = 0$, we can obtain $\gamma^{r-pf*} = \frac{8-5\alpha^2-4\beta^2}{8(2-\beta^2)}$. Substituting the optimal interest rate $\gamma^{r-pf*} = \frac{8-5\alpha^2-4\beta^2}{8(2-\beta^2)}$ into the response functions yields $w^{r-pf*} = \frac{24-5\alpha^2-12\beta^2}{6(8-3\alpha^2-4\beta^2)}$, $m^{r-pf*} = \frac{24-5\alpha^2-12\beta^2}{6(2-\beta^2)(8-3\alpha^2-4\beta^2)}$, $p^{r-pf*} = \frac{(3-\beta^2)(24-5\alpha^2-12\beta^2)}{6(2-\beta^2)(8-3\alpha^2-4\beta^2)}$, $e^{r-pf*} = \frac{4\alpha}{24-9\alpha^2-12\beta^2}$, and $t^{r-pf*} = \frac{\beta(24-5\alpha^2-12\beta^2)}{6(2-\beta^2)(8-3\alpha^2-3\beta^2)}$. Finally, by substituting $w^{r-pf*}$, $m^{r-pf*}$, $e^{r-pf*}$, $t^{r-pf*}$, and $\gamma^{r-pf*}$ into the market demand and the profits of the manufacturer, the platform, and the bank, Proposition 4 can be obtained. $\square$

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
