# Peer review of "Green R&D Financing Strategy in Platform Supply Chain with Data-Driven Marketing"

_sustainability, doi:10.3390/su14159172_

Round 1

Reviewer 1 Report

Please find the report from the attachment.

Reviewer 2 Report

Following are my comments pertaining to the paper titled "Green R&D Financing Strategy in Platform Supply Chain with Data-driven Marketing",

1. The motivation of the research work needs to be better articulated by presenting the importance of the issue from the perspective of real-world. If possible authors can establish the addressing such issues from organization's perspective. Highlight certain cases where real-world organizations are facing such issues addressed in the research work.

2. Motivation should also be drawn from the research gaps obtained from different published articles. I believe there should be a separate sub-section named research gaps focussing on the research issues obtained from the past research works. Furthermore, literature review needs to be strengthened by including relevant research works from top journals. some of the examples are given below,

Selection of best buyback strategy for original equipment manufacturer and independent remanufacturer–game theoretic approach, International Journal of Production Research

Optimization model for sustainable food supply chains: An application to Norwegian salmon, Transportation Research Part E: Logistics and Transportation Review

3. Notations used within the game model could have been presented in a table to make it easier for the readers.

4. Proofs of propositions 3 and 4 are presented, yet requires more adequate description, which is currently missing.

5. Authors can provide a managerial implication section to highlight useful insights to state the relevance of the work from practical perspective.

6. Furthermore, it is essential that you provide contribution to theory by connecting the implications from results with past literatures to clearly elaborate the ways current study is adding value to the body of knowledge.

Round 2

Reviewer 2 Report

Authors have adequately addressed the comments and now the paper can be accepted for publication.